# No effect of a dairy-based, high flavonoid pre-workout beverage on exercise-induced intestinal injury, permeability, and inflammation in recreational cyclists: A randomized controlled crossover trial

**Stephanie Kung[1], Michael N. Vakula[2], Youngwook Kim[2], Derek L. England[2], Janet Bergeson[1], Eadric Bressel[2], Michael Lefevre[1], Robert Ward[1]** *

**1** Department of Nutrition, Dietetics, and Food Sciences, Utah State University, Logan, Utah, United States of America, **2** Department of Kinesiology and Health Science, Utah State University, Logan, Utah, United States of America

* Robert.ward@usu.edu

## Abstract

### Background

Submaximal endurance exercise has been shown to cause elevated gastrointestinal permeability, injury, and inflammation, which may negatively impact athletic performance and recovery. Preclinical and some clinical studies suggest that flavonoids, a class of plant secondary metabolites, may regulate intestinal permeability and reduce chronic low-grade inflammation. Consequently, the purpose of this study was to determine the effects of supplemental flavonoid intake on intestinal health and cycling performance.

### Materials and methods

A randomized, double-blind, placebo-controlled crossover trial was conducted with 12 cyclists (8 males and 4 females). Subjects consumed a dairy milk-based, high or low flavonoid (490 or 5 mg) pre-workout beverage daily for 15 days. At the end of each intervention, a submaximal cycling trial (45 min, 70% $VO_2$max) was conducted in a controlled laboratory setting (23˚C), followed by a 15-minute maximal effort time trial during which total work and distance were determined. Plasma samples were collected pre- and post-exercise (0h, 1h, and 4h post-exercise). The primary outcome was intestinal injury, assessed by within-subject comparison of plasma intestinal fatty acid-binding protein. Prior to study start, this trial was registered at ClinicalTrials.gov (NCT03427879).

### Results

A significant time effect was observed for intestinal fatty acid binding protein and circulating cytokines (IL-6, IL-10, TNF-α). No differences were observed between the low and high flavonoid treatment for intestinal permeability or injury. The flavonoid treatment tended to

**Data Availability Statement:** All relevant data are within the paper and its Supporting Information files.

**Funding:** Funding for this project was provided by the Office of Research at Utah State University, the Utah Agricultural Experiment Station and Building University-Industry Linkages through learning and Discovery (BUILD) Dairy program program of the Western Dairy Center (Utah State University, Logan) with financial support from Dairy West (Meridian, ID) and regional processing companies. SK was supported in her PhD research by a Presidential Doctoral Research Fellowship from the Office of Research at USU, and BUILD Dairy provided a 1:1 match. Funds for the clinical trial and consumables were provided by a stipend to RW from BUILD Dairy and the Utah Agricultural Experiment Station. The funders had no role in study design, data collection and analysis, decision to publish or preparation of the manuscript.

**Competing interests:** The authors have declared that no competing interests exist.

increase cycling work output ($p = 0.051$), though no differences were observed for cadence or total distance.

## Discussion

Sub-chronic supplementation with blueberry, cocoa, and green tea in a dairy-based pre-workout beverage did not alleviate exercise-induced intestinal injury during submaximal cycling, as compared to the control beverage (dairy-milk based with low flavonoid content).

## Introduction

The gastrointestinal (GI) barrier is a selectively permeable membrane that facilitates water and nutrient uptake while excluding potentially harmful antigens and microorganisms [1]. Intestinal permeability refers to the movement of luminal contents to the internal milieu through either the intestinal epithelial cells lining the GI tract or the interstitial space between neighboring cells, which is regulated by tight junction protein complexes [1]. Both external and internal stimuli can affect intestinal permeability, and elevated intestinal permeability is a common feature of several autoimmune, intestinal, and metabolic diseases and conditions [1].

In particular, high-intensity endurance exercise causes transient, increased intestinal permeability [2,3]. During exercise, up to 80% of splanchnic blood flow can be shunted to the muscles and other peripheral tissues [4]. Prolonged hypoperfusion of visceral tissues causes localized hypoxia and mucosal injury, and reperfusion injury can occur due to reactive oxygen species (ROS) accumulation [4]. Severe or prolonged exercise-induced intestinal permeability can also cause local and systemic inflammation—potentially hindering physical performance and recovery [5]. In addition to exercise intensity, core temperature is thought to be another factor affecting exercise-induced intestinal permeability—causing epithelial cell injury and disrupting tight junctions both *in vitro* and *in vivo* [2]. A systematic review of human exercise studies by Pires et al. reported a positive association between the magnitude of core temperature change and increased intestinal permeability; increased intestinal permeability was always observed with post-exercise core temperatures above 39˚C [4].

Flavonoids are the largest subclass of compounds within polyphenols (plant secondary metabolites abundant in fruits and vegetables), and preclinical studies have demonstrated that both flavonoids and other polyphenols can regulate the intestinal barrier [6]. For example, an anthocyanin-enriched fraction isolated from highbush blueberries dose-dependently restored intestinal barrier function *in vitro* in response to an *E. coli* challenge, a model of gut barrier dysfunction [7]. In another study, a catechin-rich green tea extract attenuated high-fat diet-induced gut inflammation, permeability, and dysbiosis by modulating tight junction protein and hypoxia inducible factor-1α expression [8]. Epigallocatechin gallate, the primary flavan-3-ol in green tea, has also been shown to improve intestinal permeability, inflammation, injury, and tight junction expression in murine models [9,10]. Finally, cocoa polyphenols also have potential to improve intestinal integrity; cocoa supplementation restored colonic tight junction protein expression and reduced pro-inflammatory cytokine expression in a diabetic rat model [11].

With a few exceptions, few clinical trials have evaluated the effects of flavonoids and other polyphenols on GI permeability or exercise-induced injury. First, as intestinal permeability often increases with age, a crossover study was conducted with subjects 60 years and older comparing the effects of a high polyphenol diet (~1391 mg polyphenols per day) versus a

control diet (~812 mg polyphenols per day) [12]. The study investigators reported that the high polyphenol diet, containing polyphenol-rich foods such as cocoa, berries, and green tea, reduced serum zonulin, a surrogate marker of intestinal permeability [12]. In a study with young adults, Szymanski et al. demonstrated that 3-day supplementation with the polyphenol curcumin (500 mg/day) reduced markers of GI damage, the inflammatory response, and thermal strain following one hour of sub-maximal running in hot conditions [13]. Post-exercise intestinal injury, as determined by plasma I-FABP, was reduced with curcumin supplementation compared to placebo: 58% vs. 87% increase from pre-exercise, respectively [13]. Core temperature and heart rate were significantly lower with curcumin than placebo during the last 20 minutes of the exercise test [13]. In another study, two weeks of flavonoid supplementation (329 mg/day of quercetin, anthocyanins, and flavan-3-ols) reduced intestinal permeability both at rest and after 45 minutes of walking at ~60% VO$_2$max, but this effect was not observed following a 2.5 hour run at ~70% VO$_2$max with a group of runners [14]. The findings from these studies suggest that a dietary intervention with flavonoid-rich foods may have potential to modulate intestinal barrier function. As these flavonoids are also of interest in sports nutrition due to their vasodilatory, anti-oxidative, and anti-inflammatory effects, we developed a powdered blueberry, cocoa, and green tea pre-workout mix to evaluate its effects on the GI barrier. The purpose of this study was to determine the effects of sub-chronic (fifteen-day) consumption of this high flavonoid pre-workout beverage (HFB) on GI permeability and inflammation following one hour of high-intensity endurance exercise (45-min at 70% VO$_2$max and 15-min maximal effort time trial). Our primary outcome measure was intestinal injury, as indicated by plasma intestinal fatty acid-binding protein (I-FABP). Secondary outcomes included intestinal permeability (excreted urinary sugars), inflammatory response (circulating cytokines), and cycling performance (work output and cycling distance). Based on prior work, we hypothesized that the HFB would ameliorate exercise-induced intestinal injury and permeability relative to a low flavonoid control (LFB). Second, we also hypothesized that resilience to exercise-induced changes in GI permeability will result in a reduction in acute exercise-associated inflammation and improved cycling performance.

## Materials and methods

### Ethical approval

This study was conducted in accordance with the 2008 Helsinki Declaration for Human Research Ethics and approved by the Utah State University (USU) Institutional Review Board (IRB #9255). Participants were informed of all study requirements and provided written informed consent prior to enrollment. The protocol was registered at ClinicalTrials.gov (NCT03427879).

### Participants

Participants were recruited by word of mouth and through advertisements posted at local cycling businesses and gyms in Logan, Utah from August 2018 through February 2019; all clinic activities were concluded in May 2019. Sixty subjects were assessed for eligibility. Forty-one subjects did not meet inclusion criteria, and five did not schedule an in-person screening (Fig 1). Eligibility and general health were assessed with a participation readiness questionnaire (PAR-Q) and health screening assessment. Inclusion criteria were as follows: male or female of any race or ethnicity, age 18–55 years, cycling at least 3 times per week, participated in a cycling event in the past 12 months, and free of chronic disease and GI conditions. Exclusion criteria included a medical history of heart disease, hypertension, diabetes, Crohn's disease, IBS, colitis, celiac disease, inflammatory or autoimmune disease, allergies to pre-workout

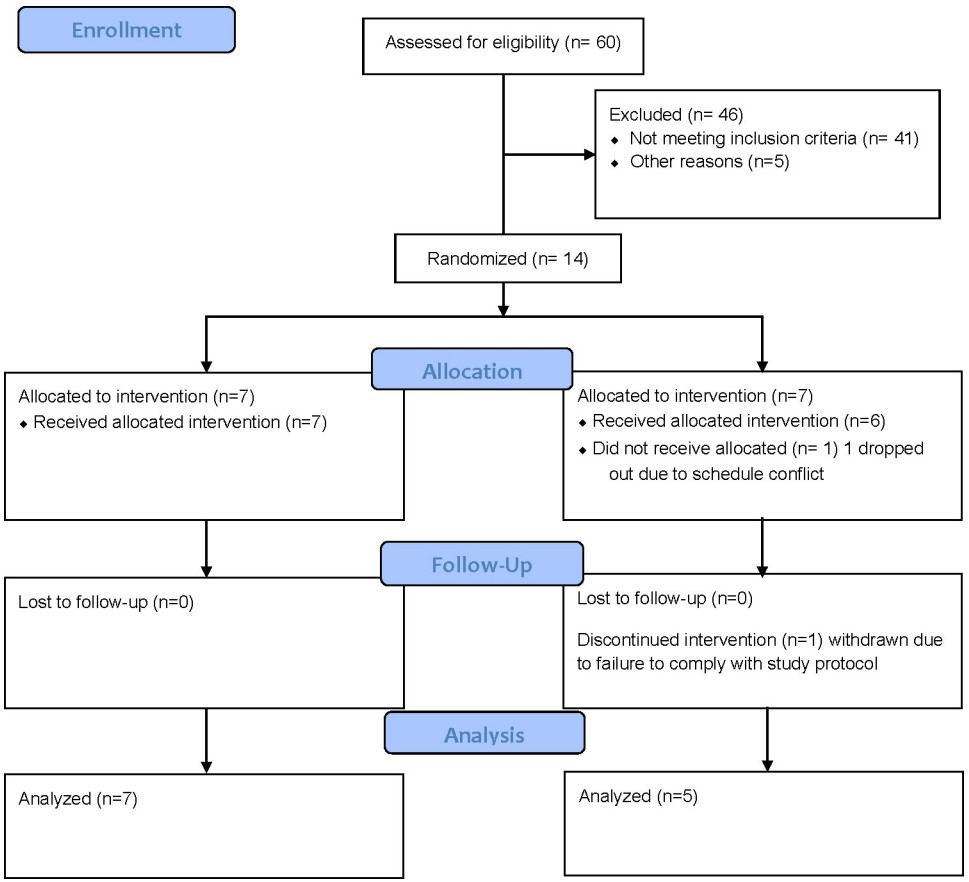

**Fig 1. Flow diagram for the study population.**

ingredients, and/or lactose intolerance. Participants were also excluded for chronic use of NSAIDs, flavonoid supplement consumption < 1 month prior, or antibiotic use < 3 months prior. For women, additional exclusion criteria included those who were pregnant, breastfeeding, or < 6 months postpartum. SK generated the random allocation sequence, enrolled the participants and assigned the participants to the interventions. Briefly, ten AB and ten BA sequences were assigned a random number with the RAND function in Microsoft Excel and ordered by ascending value. Fourteen subjects were enrolled and assigned a treatment sequence in order of enrollment. One subject dropped out due to scheduling conflicts, and one subject was withdrawn due to failure to adhere to the study protocol. Twelve cyclists (female n = 4, male n = 8) completed both arms of the study (Table 1) in the Center for Human Nutrition Studies on the Utah State University Campus in Logan, UT. Participant characteristics are included in Table 1, and subjects are classified as Performance Level 1 or untrained cyclists per De Pauw et al. [15].

## Preliminary assessments

During the screening period, each participant completed a maximal aerobic capacity ($VO_2$max) test via expired gas analysis. This preliminary test also allowed participants to be familiarized with the Velotron cycle ergometer (SRAM LLC; Chicago, IL). After a five-minute warm-up at 100 W, the incremental graded $VO_2$max test began at 100 Watts (W) and increased by 25 W/min until either volitional exhaustion, rpm < 60, or a plateau in $VO_2$ was

**Table 1. Participant characteristics.**

|  | Mean ± SD | Range |
|---|---|---|
| **Age (years)** | 36.5 ± 11.4 | 18–55 |
| **Body mass (kg)** | 77.7 ± 13.1 | 58.2–100 |
| **Height (cm)** | 174.0 ± 9.0 | 158.1–183.6 |
| **BMI (kg/m$^2$)** | 25.6 ± 3.3 | 21.8–33.2 |
| **VO$_2$max (ml/kg/min)** | 43.2 ± 5.9 | 28.1–51.4 |
| **70% Power (W)** | 160.8 ± 29.3 | 105–215 |
| **Cycling experience (years)** | 11.4 ± 6.7 | 1–23 |

observed. Expired gas samples were collected using a two-way valve mouthpiece (Hans Rudolph 700 series; Kansas City, MO), and oxygen consumption was recorded with a computerized on-line metabolic measurement system (Parvomedics TrueOne 2400; Sandy, UT). After a ten-minute rest, a follow-up assessment was performed to validate the predicted power (W) at 70% VO$_2$max during steady state exercise. Power was adjusted in 5 W increments as needed to achieve 70% VO$_2$max for a minimum of 3 minutes before the final power setting for the exercise trials was determined for each subject.

Participants also performed a baseline intestinal permeability assessment with an overnight, 8-hour urine collection. For the test, participants fasted four hours following their evening meal. After 4 hours, participants emptied their bladder and drank a 130 ml sugar solution containing 10 g sucrose, 5 g lactulose, 1 g maltitol, and 1 g sucralose. (The intended sugar permeability test includes mannitol, but due to a supplier error, maltitol was provided instead.) For the next eight hours, urine was collected in a 3 L container with 50 μl of 10% thymol as a preservative. During this time, participants were only permitted to drink water. In the morning, participants voided into the collection jug and recorded the total volume. A subsample (50 ml) was collected, and samples were stored frozen for future analysis.

## Investigational product

While previous studies with flavonoid supplementation have spanned a large range of doses (100 to 2,000 mg/day), a recent review of these studies concluded that ~300 mg flavonoids or polyphenols consumed one to two hours pre-exercise could be sufficient for supporting exercise capacity and performance [16]. Another study demonstrated an improvement in intestinal permeability with a high polyphenol diet that provided an additional 580 mg/day of polyphenols relative to the control diet [12]. As no prior clinical trials with supplemental flavonoids and exercise-induced GI injury have been conducted thus far, the total flavonoid content of the HFB was formulated to deliver a relatively moderate dose of total flavonoids (490 mg) in a single serving while maintaining a positive sensory profile. For each intervention, participants were provided with fifteen individual servings of each pre-workout mix (S1 Table) and shelf-stable white milk (2% milkfat). One package of pre-workout mix containing sucrose, maltodextrin, blueberry powder, cocoa powder, whey protein isolate and green tea (64 g) was prepared with 240 ml milk and consumed once per day, two hours prior to exercise or the subject's typical exercise time. Both the HFB and LFB were prepared in milk, which has favorable sensory and nutritional attributes and was expected to be well-liked by the subjects, and also contained some protein and fat. The low flavonoid beverage (LFB) contained approximately 4.6 mg flavonoids due to three modifications: alkalized cocoa, blueberry placebo powder, and omission of green tea powder. The green tea extract contained approximately 800 mg/g tea catechins (PureBulk, Inc; Roseburg, Oregon). High flavonoid cocoa powder (83 mg/g

cocoa flavanols) and alkalized cocoa (10 mg/g flavanols) were provided by Barry Callebaut (Zurich, Switzerland). Freeze-dried blueberry powder (14.0 mg/g total anthocyanins) and blueberry placebo powder were supplied by the US Highbush Blueberry Council (Folsom, California).

## Study endpoints

The primary endpoint of this study was I-FABP detected in the plasma, a non-invasive biomarker of intestinal injury and permeability [5]. I-FABP is a cytosolic protein expressed in mature small and large intestinal enterocytes and present only at low levels in the plasma unless released from injured or lysed cells [5]. Because circulating I-FABP is rapidly cleared and has a half-life of only eleven minutes, the change in I-FABP from baseline is an early marker for intestinal injury [17].

The second endpoint was the urinary excretion of orally-ingested sugars (sucrose, lactulose, and sucralose)—a non-invasive method to determine GI permeability. Sucrose, which is digested by the enzyme sucrase in the small intestine, is an indicator of gastric permeability, lactulose is fermented by the microbiota in the colon and therefore indicates gastric and small intestinal permeability, and sucralose is non-digestible, so recovered sucralose in the urine reflects overall GI permeability [1,18].

The plasma cytokines TNF-α, IL-6, and IL-10 were selected as secondary endpoints. The pro-inflammatory cytokine TNF-α increases the rate of epithelial cell turnover and activates opening of the tight junction, leaving gaps in the intestinal barrier [1,19]. IL-6 is secreted from skeletal muscle during exercise in response to heat stress and low energy availability [20]. It can also affect gastric emptying and plays an important role in attenuating intestinal permeability and mucosal injury caused by a dangerous rise in core temperature [21]. Lastly, IL-10 is an anti-inflammatory cytokine promoted by IL-6 that downregulates further production of pro-inflammatory cytokines [20].

## Study design

A randomized crossover trial was conducted with a minimum fourteen-day washout between the two intervention periods (Fig 2). Following completion of screening and baseline assessments, each participant was randomly assigned a treatment order. The HFB and LFB were coded, and participants were blinded to the treatment order. Study staff administering the cycling test and processing the plasma and urine specimens were also blinded to the treatment assignment.

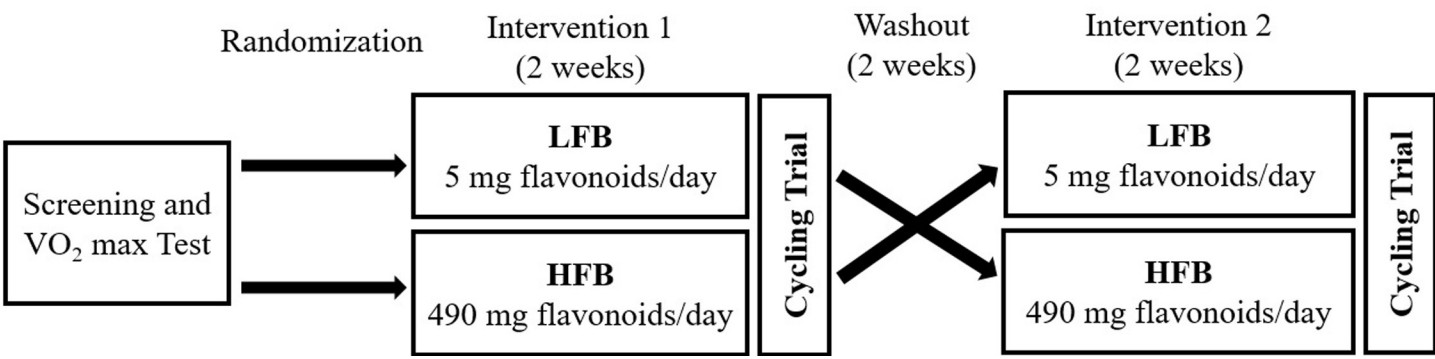

**Fig 2. Schematic overview of the crossover study design.**

With the exception of high flavonoid foods, flavonoid supplements, and NSAIDs, participants were free to consume their regular diet and maintain their usual training schedule. In addition, participants were instructed to abstain from alcohol and strenuous activity 24 hours prior to each urine collection and cycling trial. Gut permeability assessments using orally-administered sugar test probes and an overnight eight-hour urine collection were performed at baseline and on Day 12 of each intervention. On Day 15, a gut permeability assessment was also performed immediately following the trial with a six-hour urine collection. Meals consumed during these periods were recorded and replicated during the second intervention. Each intervention concluded with a one-hour cycling trial on Day 15, during which blood samples were collected pre-, post-, 1 h post-, and 4 h post-exercise.

On Day 15, two hours prior to the cycling trial, participants consumed their typical pre-race breakfast (recorded in a food log and replicated during the second arm), the HFB/LFB pre-workout, and an ingestible temperature sensor pill (CorTemp, HQ Inc.; Palmetto, FL). At the clinic, subjects emptied their bladders, body mass was measured, and a blood sample was taken. The ambient temperature of the testing room was 23°C. Subjects warmed up for five minutes at a self-selected pace prior to starting the cycling trial. The trial consisted of 45 minutes cycling at a fixed power (70% $VO_2max$) and a 15-minute "all out" time trial (TT), during which subjects cycled at a self-selected power and were instructed to ride at maximal effort. Throughout the exercise test, heart rate and core temperature were recorded at five-minute intervals, and a final core temperature measurement was taken five minutes after completion of exercise. Borg's CR10 rating of perceived exertion (RPE) was recorded at minutes 5, 15, 25, and 35 [22], and expired air was collected to determine $VO_2$ at 10, 20, 30, and 40 min. Subjects were provided with a small amount of room temperature water (0.5 ml/kg bodyweight) at 15 minutes. Following the trial, participants were seated and another blood sample was collected. Participants then ingested the gut permeability sugar solution and collected urine for six hours. During the first two hours, subjects were only permitted to drink water, and certain foods and beverages were restricted to minimize interference with the test. All meals and snacks were recorded in a food log and repeated during the second intervention. Two final blood samples were collected 1h and 4h post-exercise. Adverse events and symptoms experienced in the 24h following the exercise test were noted with brief questionnaire. Subjects were queried regarding adverse events at each clinic visit. No adverse events were reported.

## Blood sampling and analysis

Seated venous blood samples were collected via antecubital venipuncture in $K_2EDTA$-treated Vacutainer tubes and centrifuged (1500 × $g$, 10 min, 4°C). Plasma was aliquoted to 1.5-ml polypropylene microcentrifuge tubes and stored frozen at -80°C until further analysis (within 18 months of sample collection and storage).

A commercial enzyme-linked immunosorbent assay (ELISA) kit was used for determination of plasma I-FABP in duplicate (Hycult Biotech; Uden, The Netherlands). High sensitivity ELISA kits were used to analyze TNF-α and IL-10 according to the manufacturer's instructions (Invitrogen; Carlsbad, CA). Plasma IL-6 cytokine analysis was conducted with an ELISA kit according to the manufacturer's instructions (Invitrogen; Carlsbad, CA).

## Urine sugar analysis for gut permeability

Urinary concentrations of each of the sugar probes (i.e., recovery of the sugars in urine specimens) were determined with gas chromatography-mass spectroscopy (GC-MS) analysis following trimethylsilyl (TMS) derivatization. Due to the low concentrations of recovered sugars, urine samples were spiked with 7.5 ppm sucrose, 2 ppm sucralose, and 20 ppm lactulose prior

to drying and derivatization to increase the signal to noise ratio for the measurement. For the standards, a synthetic urine matrix was prepared with 0.33 M urea, 0.12 M sodium chloride, 0.016 M potassium diphosphate ($KH_2PO_4$), 0.007 M creatinine, and 0.004 M sodium mono-phosphate ($NaHPO_4$). Additionally, inositol was added to all samples as an internal standard (50 μl, 1 mg/ml). Samples were prepared in duplicate, and 400 μl of urine or 800 μl of a 1:1 standard and synthethic urine mix were evaporated to dryness in a speed vacuum centrifuge. After drying, 100 μl of *O*-methoxylamine ($CH_3ONH_2$) in pyridine (20.0 mg/mL) was added to each sample and mixed thoroughly. Samples were heated at 70˚C for 1 hour. After heating, samples were centrifuged at 13,000 x *g* for 5 minutes, and 50 μl of the supernatant was combined with 50 μl of *N*,*O*-Bis(trimethylsilyl)trifluoroacetamide (BSTFA). The samples were heated again for 30 minutes at 70˚C and then transferred to a glass sample vial for GC-MS analysis.

The GC-MS platform was a Shimadzu single quadrupole gas chromatograph mass spectrometer (GCMS-QP2010 SE) with a Zebron ZB-5MS plus (35 m × 0.25 mm diameter × 0.25 μm film thickness) capillary GC column (Phenomenex, Torrance, CA, USA). Helium was used as the carrier gas, and the column flow rate was 1.16 ml/min, with a split ratio of 5:1. The temperatures of the inlet, ion source, and transfer line were 250˚C, 230˚C, and 290˚C, respectively. Compound peak areas were normalized by the inositol internal standard, and sugar concentrations were determined from standard curves. Recovered sugar concentrations were corrected to the actual concentration by subtracting the spiked concentration. Corrected concentrations were multiplied by total urine volume to determine recovered mass which was used to determine percent urinary recovery from the initial amount of sugar ingested.

## Sample size

A previous study demonstrated a treatment difference (dietary intervention versus placebo) of 328.6 ng/ml for the change in I-FABP following exercise, with subject standard deviation for repeated measurements estimated at 389.8 ng/ml [23]. Assuming a similar effect size of 0.84 (α = 0.05), fourteen subjects are needed to achieve 82% power for this crossover study design. If a similarly large effect size (Cohen's *d* > 0.8) is found in the present study, it would suggest that a statistically significant difference is also of practical significance, as it reflects a relatively larger minimal detectable difference relative to the variation in the I-FABP measurement.

## Statistical analysis

All data were checked for normality and homogeneity of variance, and data violating model assumptions were log-transformed prior to analysis. The TT distance, work, and cycling cadence were analyzed with paired t-tests. Cohen's d (*d*) was used to determine effect size for cycling work output, and a *d* value of 0.2, 0.5, and 0.8 indicate small, medium, and large effect sizes, respectively. Core temperature, heart rate (HR), VO$_2$, RPE, plasma I-FABP, IL-10, IL-6, and TNF-α were analyzed using a two-way repeated measures ANOVA. For IL-6 and TNF-α, statistical analyses were performed on raw instrumental values. When main or interaction (intervention × time) effects were significant, Tukey's HSD was used for post-hoc comparisons, unless otherwise specified. Negative values for sugar recovery were input as 0. A one-way repeated measures ANOVA was used to analyze sugar probe recovery with condition (baseline, HFB Day 12, LFB Day 12) as the independent factor. Intestinal permeability following the exercise trial on Day 15 was analyzed with a paired t-test. Correlation analyses (Spearman's ρ) were conducted to determine if final core temperature and change in core temperature correlated with primary outcomes (change in I-FABP, post-exercise I-FABP, % sucrose, % lactulose, % sucralose). No significant treatment order effect (HFB-LFB, LFB-HFB) was observed. Data

are expressed as means ± SEM, unless otherwise indicated. Statistical analyses were performed with JMP version 15.2.1 for Windows (SAS Institute, Cary, NC) with significance at α = 0.05.

## Results

### Physiological responses and exercise performance

The exercise test elicited a similar response in HR, RPE, and $VO_2$ regardless of the treatment (S1 Fig and S2 and S3 Tables). There was a significant effect of exercise on HR ($p < 0.0001$), where the HR during the TT (t = 50, 55, 60) was significantly higher than the pre-load exercise (t = 5–45). Likewise, RPE (scale of 0–10) increased during exercise and was significantly elevated at 25 and 35 min compared to 5 min ($p < 0.0001$). Final RPE at 35 min was 3.9 ± 0.5 (LFB) and 4.0 ± 0.4 (HFB), where an RPE of 4 is "somewhat hard". $VO_2$ was significantly lower at 10 min compared to all other times ($p = 0.0003$), but not different between trials.

No differences were observed for TT cadence (LFB: 93 ± 11 rpm, HFB: 93 ± 9 rpm, p = 0.938) or TT cycling distance (LFB: 7.3 ± 0.3 km, HFB: 7.4 ± 0.3 km, p = 0.546) (Fig 3A and 3B). However, a trend for increased TT work output was observed for the HFB, with a within-subject treatment difference of 6.6 ± 3.0 kJ (p = 0.0511, Cohen's $d$ = 0.16) (Fig 3C).

A significant effect of time ($p < 0.0001$) and time × treatment interaction ($p < 0.0001$) was observed for core temperature (Fig 4). Core temperature from 10 min onwards for both conditions was significantly elevated compared to pre-exercise. Pairwise comparisons across condition at each time were not significantly different after correcting for multiple comparisons. A small decline in core temperature was observed at 20 min when participants were provided water (0.5 ml water/kg bodyweight) after 15 min of exercise. The maximum core temperatures immediately following exercise were 38.53 ± 0.06˚C (HFB) and 38.51 ± 0.23˚C (LFB). Overall changes in core temperature from pre-exercise (t = -5) to 60 min exercise were 2.24 ± 0.23˚C (LFB) and 1.73 ± 0.13˚C (HFB) (paired t-test, p = 0.008).

### Inflammatory markers

A main effect of time was observed for plasma IL-6 and IL-10 where both cytokines were significantly elevated post-exercise and after 1-h recovery (Table 2). Similarly, TNF-α was elevated immediately post-exercise compared to pre-exercise (Table 2). No significant effects were observed for treatment or time × treatment interaction.

### Gut injury and permeability

A significant effect of time was observed for plasma I-FABP ($p = 0.001$), but there was no significant treatment effect or treatment × exercise interaction (Fig 5). Plasma I-FABP was significantly elevated immediately post-exercise compared to all other time points. Mean change in I-FABP from pre- to immediately post-exercise did not differ between treatments: 300 ± 156 pg/ml (HFB) and 439 ± 162 (LFB) pg/ml. In addition, no significant correlation was found between final core temperature or change in core temperature and change in I-FABP.

Percent recovery of 8-h urinary sugars (sucrose, lactulose, and sucralose) at rest was not significantly different between conditions (baseline, LFB, HFB) (Table 3). Similarly, percent recovery of 6-h urinary sugars was not significantly different between treatments following the exercise trial (Table 4). Since a post-exercise sample was missing for one subject, these analyses were conducted with n = 11. No significant correlation was found between final core temperature or change in core temperature and percent recovery of sucrose, lactulose, or sucralose.

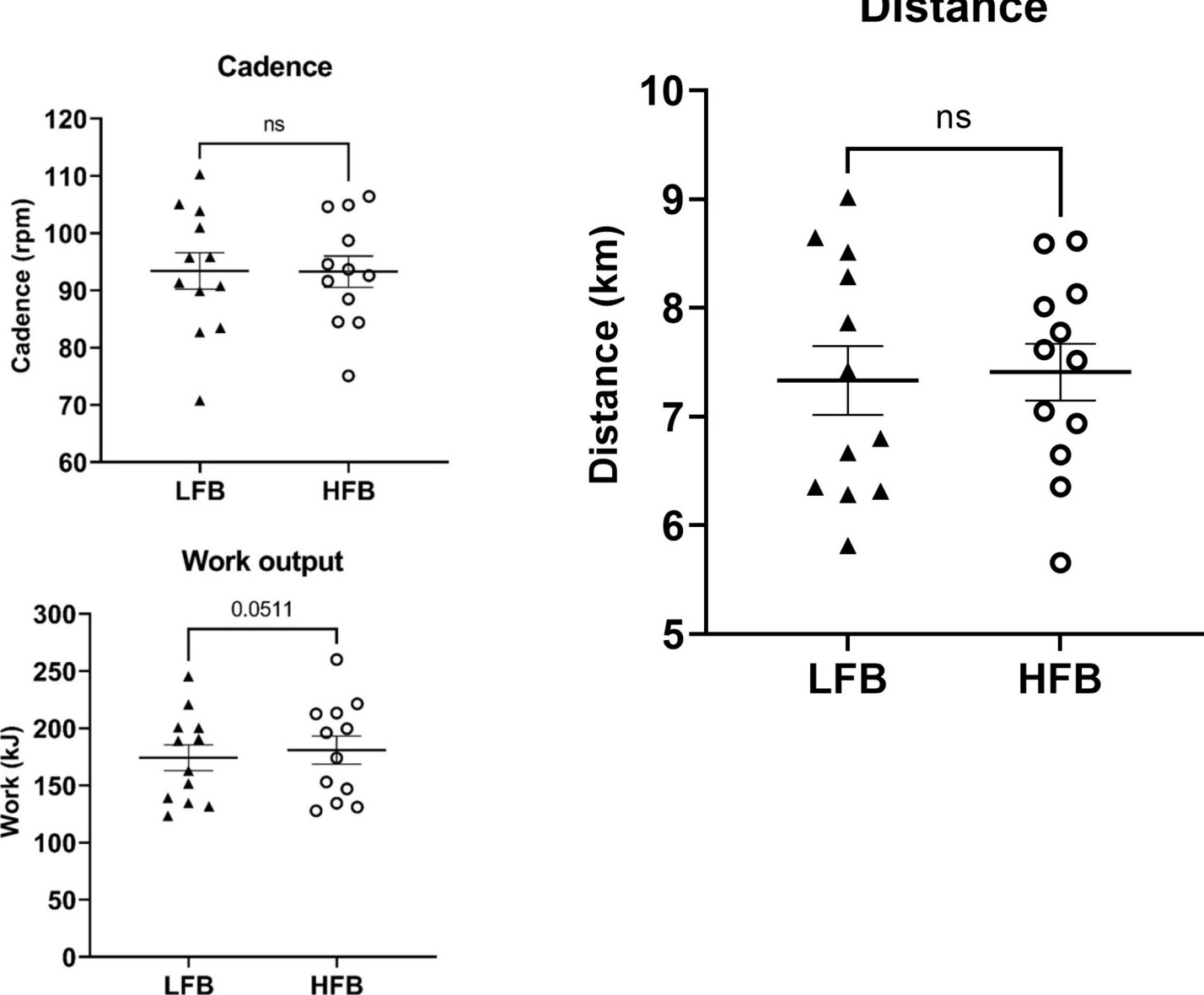

**Fig 3. Cycling performance metrics during 15-min TT.** (A) Mean cycling cadence (LFB: 93 ± 11 rpm, HFB: 93 ± 9 rpm). (B) Distance completed (LFB: 7.3 ± 0.3 km, HFB: 7.4 ± 0.3 km). (C) Work output (LFB: 174.2 ± 11.3 kJ, HFB: 180.9 ± 12.3 kJ).

## Discussion

The purpose of the present study was to determine the effect of flavonoid supplementation on intestinal inflammation, injury, and permeability following cycling at 70% VO$_2$max. During the HFB intervention, cyclists consumed ~490 mg/day of cocoa, blueberry, and green tea flavonoids in a milk-based pre-workout beverage for two weeks leading up to the cycling trial. The 1-hour cycling trials for both conditions were consistent in degree of exercise intensity, with similar responses in RPE, VO$_2$, HR, and peak core temperature. Contrary to our hypothesis, we observed no flavonoid treatment effects on GI permeability, injury, or inflammation.

Though there were no difference in cycling distance or cadence, HFB tended to increase cycling work during the TT. Further post-hoc analyses showed that cycling power also tended

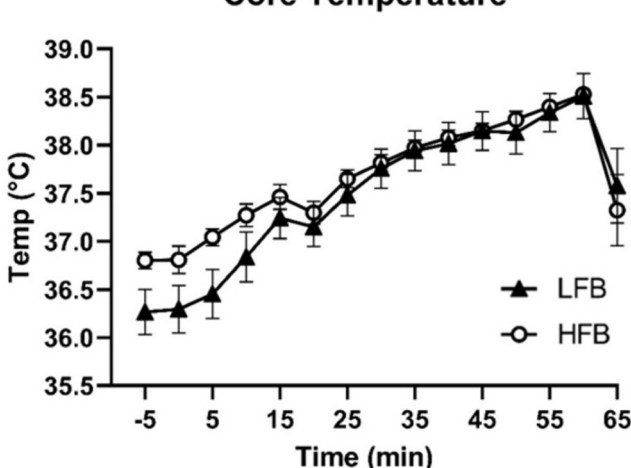

**Fig 4. Core temperature during the 1h cycling trial.**

to be greater in the HFB trial (HFB: 201 ± 14 W and LFB: 194 ± 13 W, p = 0.051). Cadence was not different between conditions. Thus, the greater power for the HFB trial was likely the result of participants applying more force to the pedals. Given the achieved sample size and an estimated within-subject standard deviation of 9.3 kJ for repeated TT measurements [24], post-hoc power analysis ($\alpha = 0.05$, $\beta = 0.80$) showed that the study was powered to detect a minimum treatment difference of 11.7 kJ. However, the small effect size ($d = 0.16$) suggests that the observed difference in work (6.6 kJ) is not likely to have a meaningful impact on overall performance—which is supported by lack of an effect on total distance (HFB: 7.4 ± 0.3 km and LFB: 7.3 ± 0.3 km).

Other studies have found performance benefits with a variety of other dietary flavonoids, which, as discussed in a recent review by Bowtell and Kelly, can activate endogenous antioxidant pathways via Nrf2 signaling to reduce oxidative stress and improve muscle perfusion through increased NO availability [16]. Presumably, this would have an effect on performance, perhaps through delaying the onset of fatigue. Several studies with both acute and chronic

**Table 2. Plasma cytokines pre-and post-exercise.**

| | | Pre-exercise | Post-exercise | | |
|---|---|---|---|---|---|
| | | | **0h** | **1h** | **4h** |
| IL-6 (pg/ml) | HFB | n.d. | 0.76 ± 0.23* | 0.22 ± 0.13* | n.d. |
| | LFB | n.d. | 0.79 ± 0.38* | 0.53 ± 0.20* | n.d. |
| | HFB-LFB | n.d. | -0.03 ± 0.42 | -0.31 ± 0.19 | n.d. |
| IL-10 (pg/ml) | HFB | 1.49 ± 0.36 | 3.95 ± 0.69* | 3.04 ± 0.44* | 1.48 ± 0.28 |
| | LFB | 1.52 ± 0.38 | 3.76 ± 0.67* | 2.69 ± 0.36* | 1.86 ± 0.55 |
| | HFB-LFB | -0.02 ± 0.36 | 0.20 ± 0.69 | 0.35 ± 0.44 | -0.39 ± 0.28 |
| TNF-α (pg/ml) | HFB | n.d. | 0.28 ± 0.09* | n.d. | n.d. |
| | LFB | n.d. | 0.31 ± 0.07* | n.d. | n.d. |
| | HFB-LFB | n.d. | -0.02 ± -0.06 | n.d. | n.d. |

Mixed effects, repeated measures 2-way ANOVA performed on raw instrumental values. n.d., not detected.

*Significant difference from pre-exercise, $p < 0.0001$.

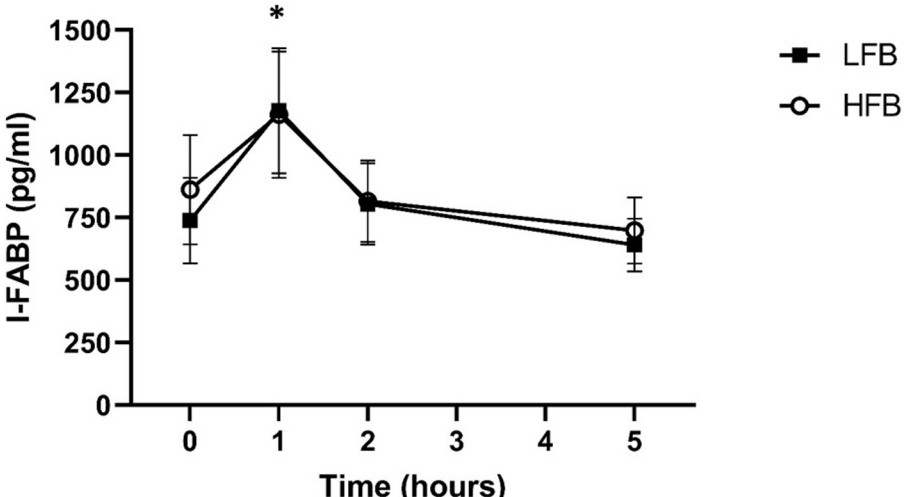

**Fig 5. No treatment effect on plasma I-FABP pre- or post-exercise.** *Significant difference from pre-exercise, p < 0.05.

blackcurrant, pomegranate, and cherry anthocyanin supplementation have demonstrated improved performance in both trained and untrained subjects [25–29]. Both acute and chronic cocoa supplementation have been found to reduce oxidative stress, though improvements in exercise performance and recovery are less consistent [30–32]. Based on these studies, it is possible that we would have seen an attenuation in oxidative stress or improvement in tissue oxygenation with HFB, which would help explain the slight performance benefits observed. However, we did not measure markers of oxidative stress as this was beyond the scope of our study. Additionally, if the effects of flavonoid intake require a longer time to show measurable changes, then an improvement in performance may have been observed with a longer supplementation period.

Physical exertion and hyperthermia are known to cause changes in plasma cytokine levels, but we observed relatively low levels of IL-6, IL-10, and TNF-α despite the rigorous exercise protocol [19,33]. As expected, plasma I-FABP and IL-6 increased in response to exercise and returned to baseline levels during the recovery period, regardless of the treatment. However, plasma IL-6 and TNF-α were relatively low even immediately following exercise and mostly undetectable at baseline and after 4 hours of recovery. Similarly, Osborne et al. found that I-FABP increased 140% from pre- to post-exercise, but no change in circulating pro-inflammatory cytokines or endotoxin following a 60-min trial at 35˚C, with final core temperatures reaching 39.5˚C [34]. Other studies at a similar intensity reported no changes in TNF-α (< 2 pg/ml pre- to post-exercise) and plasma cytokines IL-10 and IL-6 (< 6 pg/ml) despite showing a significant increase in intestinal permeability and core temperature of 39.5˚C, respectively

**Table 3. Sugar probe excretion at rest.**

|  | **Baseline** | **HFB (Day 12)** | **LFB (Day 12)** | **p-value** |
|---|---|---|---|---|
| % Sucrose | 0.035 ± 0.010 | 0.069 ± 0.023 | 0.056 ± 0.018 | 0.454 |
| % Lactulose | 0.424 ± 0.087 | 0.424 ± 0.081 | 0.324 ± 0.064 | 0.354 |
| % Sucralose | 1.206 ± 0.210 | 1.311 ± 0.264 | 1.237 ± 0.448 | 0.561 |

8h overnight urine collection, n = 12

**Table 4. Sugar probe excretion post-exercise.**

|  | HFB | LFB | HFB-LFB | p-value |
|---|---|---|---|---|
| % Sucrose | 0.040 ± 0.008 | 0.066 ± 0.026 | -0.026 ± 0.028 | 0.817 |
| % Lactulose | 0.293 ± 0.050 | 0.267 ± 0.085 | 0.026 ± 0.085 | 0.384 |
| % Sucralose | 0.775 ± 0.060 | 1.116 ± 0.386 | -0.341 ± 0.404 | 0.994 |

6h post-exercise urine collection, n = 11.

[35,36]. Short-term curcumin supplementation reduced the increase in I-FABP relative to placebo following a one-hour treadmill run at 65% $VO_2$max at 37˚C, but no significant interaction effects were observed between the treatment and time for IL-6, IL-10, and TNF-α [13]. In contrast, other studies have shown exercise effects on circulating cytokines, with varying effects of polyphenol ingestion. Two-week supplementation with the flavonoids quercetin and EGCG significantly reduced IL-6 and IL-10 but not TNF-α relative to placebo following a daily three-hour cycling protocol for three consecutive days [37]. McAnulty et al. found that six weeks of blueberry ingestion elevated the anti-inflammatory cytokine IL-10 to a greater extent than the control immediately following a 2.5 h run at ~72% $VO_2$max [38]. Based on these varied results, it is likely that shorter duration exercise at temperate conditions, such as in our study, does not elicit as large of a cytokine response as is more often observed in hotter environmental temperatures, longer events, or non-controlled environments, such as a road marathon [39].

We observed a larger overall increase in core temperature for LFB due to a lower initial temperature, regardless of treatment sequence. To our knowledge, this difference in initial core temperature potentially due to two-week flavonoid supplementation has not been previously reported, but previous studies suggest that green tea, epicatechin, and chocolate supplementation may stimulate mitochondrial biogenesis and increase energy expenditure to affect thermogenesis [40–44]. Nonetheless, we do not believe that this difference is a significant concern in the final outcome, because the peak core temperatures were not different. Previous studies have demonstrated that peak core temperature is positively associated with exercise-induced GI injury and permeability [3,4]. In a systematic review, increased intestinal permeability was reported in ~36% of participants with a peak core temperature of 38.5˚C or below and in ~48% of participants from 38.6–39.0˚C [4]. Though the cycling protocol employed in our study was comparable in intensity to previous studies, the average final core temperature was relatively lower (38.52 ± 0.12˚C). Furthermore, when subjects were grouped into either low or high final core temperature (< or > 38.61˚C, respectively), no effect of low or high core temperature was found for overall change in I-FABP or percent sugar recovery. Additionally, no correlations were found between final core temperature and primary outcomes.

A post-hoc power calculation was conducted for the post-exercise I-FABP response given the achieved sample size and estimated within-subject standard deviation (380.8 pg/ml), and this study was powered to detect a treatment difference of ~480 pg/ml. The observed treatment difference was 16 pg/ml—much smaller than the variation observed within subjects and not likely a clinically meaningful difference had there been a true effect. In comparison, previous studies have reported post-exercise treatment differences of ~300 pg/ml (curcumin supplement vs. placebo), 424 pg/ml (35 vs. 20˚C environmental temperature), 328.6 pg/ml (bovine colostrum vs. control), 401 pg/ml (ibuprofen vs. control), and 186.8 pg/ml (carbohydrate gel vs. placebo) [13,23,34,45,46]. Moreover, exercise-induced intestinal injury (pre- to post-exercise change in I-FABP) was relatively modest and transient regardless of the treatment. First, plasma I-FABP was elevated immediately post-exercise but not one or four hours later.

Second, an average increase of 46% from pre- to post-exercise was observed for I-FABP, whereas other moderate- and high-intensity protocols have been shown to increase I-FABP by 50–250% [47]. Perhaps with greater exercise-induced stress (indicated by a larger elevation in I-FABP and/or final core temperature), we may have observed an effect of the flavonoid treatment on intestinal permeability and injury.

Another reason why no effects were observed may be due to the study population. Recently, Keirns et al. have suggested that chronic exercise may improve resting GI integrity over time, similar to how regular exercise can result in adaptive mechanisms in other biological systems [48]. If this is the case, trained subjects may be more resilient to changes in GI permeability than untrained subjects. However, the currently available data is limited to individuals with various metabolic conditions (insulin resistance, type II diabetes, and obesity), and these studies also reported reductions in visceral fat mass and body weight, which could improve intestinal permeability through reducing circulating inflammatory cytokines [48]. Others have suggested that flavonoid supplementation may have a greater effect in untrained subjects, since competitive athletes have more active endogenous antioxidant and anti-inflammatory responses and increased muscular efficiency [49,50]. For example, a fruit and vegetable flavonoid powder fed to highly trained cyclists for seventeen days did not have an effect on exercise-induced inflammation, but a similar dose of flavonoids was previously shown to have an effect in non-athletes [51]. In another study, two-week flavonoid supplementation decreased intestinal permeability in an untrained population both at rest and after walking 1-h at ~62% $VO_2max$, while intestinal permeability was elevated in a group of trained runners following 2.5-h running at 69% $VO_2max$—though several other factors may have also contributed to these results [14]. Overall, it is still unclear what effects regular training may have on intestinal permeability in healthy adults. While we may have observed an effect with a different subject population, the purpose of the study was to determine the effects of flavonoid intake on exercise-induced GI injury. Thus, we recruited subjects with previous cycling experience and who were training regularly. In addition, untrained subjects were less likely to successfully complete the challenging cycling trial.

An additional reason why no treatment effects were observed may be due to flavonoid bioavailability and bioactivity. Bioavailability can vary widely from purified compounds, to extracts, to fresh whole foods [52]. In a previous study, dietary flavonoids were provided through a whole food diet, while the investigational product in this study was prepared from a mixture of powdered, freeze-dried, and extracted food ingredients [53]. Although the flavonoid content of the blueberry, cocoa, and green tea powders were supplied by the manufacturers, some bioactivity may have been reduced during transit, storage, or any time prior to consumption.

Polyphenol bioavailability can also be impacted by the formation of polyphenol–protein complexes with exogenous proteins (such as those found in milk), but these noncovalent protein–polyphenol interactions are generally more likely to form with larger molecular weight polyphenols such as tannins rather than the lower molecular weight flavonoids [54]. Conflicting studies also exist regarding the extent of the effect of milk protein on flavonoid bioavailability, with some reporting no effect while others have reported lower levels of polyphenolic metabolites or lower antioxidant activity, suggesting lower bioavailability [55–60]. Interestingly, in the study by Reddy et al., the addition of milk lowered the apparent bioavailability of tea catechins by ~20% (determined by the area under the curve of catechin plasma metabolites), but did not limit its antioxidant activity *in vivo* [58]. A reason may be that the absorption of the tea catechins was slower due to the presence of milk, and the complete metabolite profile was not captured in the relatively short 3-h sample collection time frame. Another reason may be that the formation of a protein and polyphenol complex might not actually inhibit bioactivity. Polyphenols are known to have low bioavailability (~5–15%), but their bioactivity is

thought to be primarily the result of gut microbial metabolism, where their low molecular weight derivatives can then act locally or be absorbed to exert systemic effects [61]. Thus, the formation of a protein–polyphenol complex may actually protect the polyphenol through the unstable GI environment, and studies have demonstrated that these complexes can be used as a delivery vehicle for polyphenols in the development of functional foods [62–64].

A limitation of this study was that there was no monosaccharide available for the intestinal permeability assessment. The urinary excretion ratio (the recovery of the larger probe relative to the smaller) is most often reported and provides a standardized measure of intestinal permeability at the region of the GI tract where the larger probe is absorbed [17]. Due to a supplier error, a monosaccharide probe was not included in the intestinal permeability assessment, but the crossover design of the study still allowed for meaningful within-subjects comparisons for percent recovery of the disaccharide probes, and some other studies have similarly reported percent recovery of disaccharide sugars [3,65–70].

Another limitation was that the study design restricted comparisons between different urine collections. The sugar excretion test is ideally performed fasted to minimize the contribution of dietary sugars, so urine samples collected at baseline and mid-intervention were fasted, eight-hour overnight collections, but the collection immediately following the cycling trial was a six-hour collection (two hours fasted) in order to minimize participant burden and shorten the clinic visit on Day 15. As a result, comparisons for intestinal permeability are only made among baseline and mid-intervention samples and between post-exercise samples.

In conclusion, a randomized controlled crossover study was conducted to test the effects of a pre-workout high flavonoid beverage in reducing exercise-induced GI permeability and injury. During a one-hour cycling test and TT, the HFB performed as well as a standard pre-workout mix. Despite the accumulating evidence suggesting potential for improvement of GI permeability, inflammation, and athletic performance with supplemental flavonoid intake, we found no differences between the treatment and placebo. Though there was potentially an effect of increased cycling work, the difference in means relative to error was small, and no difference in TT distance was observed. Due to the importance of addressing epithelial integrity and inflammation in not only exercise but also exertional heat stress and inflammatory diseases (e.g., inflammatory bowel disease, ulcerative colitis, type I diabetes), more work—especially well-controlled human clinical trials—is warranted. Future studies should consider including measurements of tissue perfusion or vascular effects (e.g., flow mediated dilation, tissue oxygenation index), mitochondrial metabolism (e.g., mitochondrial density, citrate synthase), and oxidative stress (e.g., glutathione, protein carbonyls, total antioxidant status).

## Supporting information

**S1 Checklist.**
(DOC)

**S1 Fig. Heart rate response during the cycling trial.**
(TIF)

**S1 Table. High flavonoid beverage (HFB) and low flavonoid beverage (LFB) components.**
(DOCX)

**S2 Table. Rating of perceived exertion (RPE) during cycling.**
(DOCX)

**S3 Table. VO$_2$ (ml/kg/min) during cycling trial.**
(DOCX)

**S1 Dataset.**
(XLSX)

## Acknowledgments

The authors acknowledge Braden Harris for his assistance in the clinic and Kyle McCarty and Kaitlyn Kauzor for their assistance with phlebotomy.

## Author Contributions

**Conceptualization:** Stephanie Kung, Eadric Bressel, Michael Lefevre, Robert Ward.

**Formal analysis:** Stephanie Kung, Robert Ward.

**Investigation:** Stephanie Kung, Michael N. Vakula, Youngwook Kim, Derek L. England, Janet Bergeson, Robert Ward.

**Methodology:** Stephanie Kung, Michael N. Vakula, Youngwook Kim, Eadric Bressel, Robert Ward.

**Project administration:** Stephanie Kung, Robert Ward.

**Resources:** Eadric Bressel, Michael Lefevre.

**Supervision:** Janet Bergeson, Eadric Bressel.

**Visualization:** Stephanie Kung, Robert Ward.

**Writing – original draft:** Stephanie Kung.

**Writing – review & editing:** Michael N. Vakula, Youngwook Kim, Derek L. England, Eadric Bressel, Michael Lefevre, Robert Ward.

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
