## [Decision Letter · Decision Letter 0]

10 May 2022

PONE-D-22-01325No effect of flavonoid supplementation on exercise-induced intestinal injury, permeability, and inflammation in recreational cyclists: A randomized controlled crossover trialPLOS ONE

Dear Dr. Ward,

Thank you for submitting your manuscript to PLOS ONE. After careful consideration, we feel that it has merit but does not fully meet PLOS ONE’s publication criteria as it currently stands. Therefore, we invite you to submit a revised version of the manuscript that addresses the points raised during the review process. Please be aware that this invitation does not guarantee any future acceptance as this will be based upon your ability to rebuttal and revise based on the comments provided. Apart from the comments provided by the reviewers (see below for full list), I would urge you to describe/explain any deviations from the protocol as provided in the supplemental files and/or the trial registry, and how these changes affected the outcomes (to be integrated in discussion). Specifically, the much lower sample size (from n=20 to n=12), and the inclusion of rather untrained participants (based on participant characteristics, and unlike the suggested VO2max of >45 ml/kg/min, also age range is different). Also, for the power issue, the reasoning for taking 328 as a relevant difference is unclear (the number does not appear in the referred paper). As per the reviewers' comments, several methodological issues need to be explained and their implications discussed in the discussion. In doing so, please try to reduce length of the discussion (7 pages is too long) by focusing more on the current work, rather than extensively discussing previous studies. Finally, full supplement details should be provided, i.e., what was in the 64 g of supplement?

We look forward to receiving your revised manuscript.

Kind regards,

Lex Verdijk, PhD

Academic Editor

PLOS ONE

Journal Requirements:

4. Please include a caption for figure 5

5. Please ensure that you refer to Figure 5 in your text as, if accepted, production will need this reference to link the reader to the figure.

Reviewers' comments:

Reviewer's Responses to Questions

**Comments to the Author**

1. Is the manuscript technically sound, and do the data support the conclusions?

Reviewer #1: Yes

Reviewer #2: No

Reviewer #3: Yes

2. Has the statistical analysis been performed appropriately and rigorously? 

Reviewer #1: Yes

Reviewer #2: Yes

Reviewer #3: Yes

3. Have the authors made all data underlying the findings in their manuscript fully available?

Reviewer #1: Yes

Reviewer #2: Yes

Reviewer #3: Yes

4. Is the manuscript presented in an intelligible fashion and written in standard English?

Reviewer #1: Yes

Reviewer #2: Yes

Reviewer #3: Yes

5. Review Comments to the Author

Reviewer #1: 

I like the manuscript and the data analysis seems to be professionally done. My concern is the very tiny sample size, which is computed to be 14, in contradiction to the included protocol, which specifies 20. There is a post-hoc power analysis, which is always appropriate for a PLOS-ONE manuscript, but rarely included, so that is a positive aspect of this paper. There needs to be more detail about what the effect size actually means and how the sample size incorporated the crossover design. In addition, power is not defined correctly: "the probability is 82% that a significant treatment effect will be detected". You could just say "to achieve 82% power".

With such a small sample size, it calls into question the additional analyses besides the primary, and presumably you would want to adjust the alpha for multiplicities which will reduce the power even more.

Finally, "simple randomization" usually called "complete randomization" is, I guess, tossing a fair coin, which does not ensure equal group sizes, but I couldn't find the group sizes anyway.

Reviewer #2: 

It is not clear from the title and abstract that the supplementation was a flavonoid/milk drink. Please clarify. This also needs to be discussed. There is no mention of milk in the discussion. Could it be that milk affected digestion, absorption and availability of flavonoid-derived metabolites. Please discuss.

It is a concern that baseline IL-6 and TNF-alpha was not detectable, and also not at 4 hr post. This needs to be discussed.

In addition, observations 1 hr post were very low. Why is this.

The study did not have a full familiarization for the 1 hr cycling test. This needs to be discussed.

It is not clear what the cycling distance was during the 15 min TT.

On several occasions, statements are supported with review papers. I suggest to clarify that it is a review paper or even better provide original sources.

Please clarify in the abstract whether the 15 min of cycling was subsequent to the 1 hr of cycling, see L112.

In the discussion section of the abstract, it seems the two statement provide similar information. In addition, I suggest to remove the word “significantly” and rephrase “performed as well” suggesting as this suggests a control effect.

In the introduction there is on many occasions mention of polyphenol effects. I suggest to be specific what kind of polyphenols were providing the effect.

In the introduction, there is no mention of the flavonoid composition of the supplement of interest and why that composition would have an effect on GI health. The introduction could be strengthened by incorporating studies with effects observed by the flavonoid components of the supplement.

L149. The incremental cycling test started at 100 W and increased 25W/min. For a participant, the power at 70% was 105 W suggesting that this person and maybe more did only a few minutes of cycling to reach VO2max. Please clarify.

L169. Was 2% the fat content. Please clarify.

L170. The supplement was dissolved in milk so there was protein intake as well and not only flavonoids. Please clarify.

Ls 175-179. Please provide justification for the amounts of green tea extract, cocao and blueberry.

Ls 216-217. What was the composition of the “typical pre-race breakfast”. Why was this not standardized. This needs to be discussed.

Ls 325. Please provide pre-exercise IL-6 value to allow the statement that there was an increase post-exercise.

L 372. Mention of blackcurrant and pomegranate studies and then the subsequent statement provide observations from a cherry study whereas the “For example,…” would suggest an example of a blackcurrant or pomegranate study.

L395. Why were some so low (and even non-detectable). Was there an issue with the assay?

Reviewer #3: 

This paper examines the effects of 15 days of flavonoid supplementation on gastrointestinal injury, permeability and inflammation after a one-hour cycling trial. The paper is well written and has experimental design strengths. Nevertheless, there are areas for clarification and improvement.

Line 64: “high intensity” is missing a hyphen. Please add.

Line 73: A determiner is missing before “magnitude”. Please add an article.

Line 85: Please a comma before and after “as determined by plasma I-FABP”.

Line 93: The preposition use may be incorrect here. Please consider if “at” should be ‘on’.

Line 100: A determiner is missing before “onset”. Please add an article.

Line 113: Please consider if “primary endpoint” is the best word choice. ‘primary outcome measure’ might be clearer.

Line 113: A comma is missing after “injury”. Please add.

Line 116: Intestinal damage is most often higher in runners than cyclists. With this in mind, why was cycling chosen as the exercise model?

Line 124: Please add the ethics approval number.

Line 130: Please reference the tool used to complete the randomization.

Line 135-6: For ease of comparison with other studies, please consider adopting the descriptors of De Pauw et al.

Reference: De Pauw, K., Roelands, B., Cheung, S. S., De Geus, B., Rietjens, G., & Meeusen, R. (2013). Guidelines to classify subject groups in sport-science research. International Journal of Sports Physiology and Performance, 8(2), 111-122.

Line 143: Te range in participant characteristics is large. Does this heterogeneity concern the authors?

Line 150: How was a plateau defined?

Line 170-1: What does the subjects’ typical exercise look like? These details are not included in S1 Table.

Line 171-2: Were the supplements taste matched? Were participants able to detect whether they were in the high or low condition? Does this matter?

Line 217-8: Calibration of core temperature pills is ideal (Hunt et al., 2017). Was this undertaken here?

Reference: Hunt, A. P., Bach, A. J., Borg, D. N., Costello, J. T., & Stewart, I. B. (2017). The systematic bias of ingestible core temperature sensors requires a correction by linear regression. Frontiers in Physiology, 8, 260.

Line 218-9: Was hydration examined via either urine or blood?

Line 228-9: Please specifically identify the RPE scale used (e.g., Borg’s 6-20 or CR10?). Other details including familiarization and anchoring (Haile, Gallagher & Robertson, 2016) and the standardization of the recording context (Minett et al., 2021), would also be worthwhile.

References: Haile, L., Gallagher, M. J., & Robertson, R. J. (2016). Perceived exertion laboratory manual. Springer-Verlag New York.

Minett, G. M., Fels-Camilleri, V., Bon, J. J., Impellizzeri, F. M., & Borg, D. N. (2021). Peer presence increases session ratings of perceived exertion. International Journal of Sports Physiology and Performance, 17(1), 106-110.

Line 238: Could the authors please detail the environmental conditions?

Line 242: How long were samples stored before analysis?

Line 247: Please describe the reliability of these measures.

Line 273: As per the immediately previous comment.

Line 279: Is it appropriate to combine male and female data in this context?

Line 295: Could the authors justify the presentation of SEM and not SD values?

Line 318-9: The decline in core temperate after fluid consumption suggests the pills were still in the stomach. This should be acknowledged as stomach temperature, not gastrointestinal temperature, is being reported.

Line 342: “condition” should be in the plural form. Please correct.

Line 344-5: Did the authors consider using a mixed-method or similar instead of an ANOVA to accommodate this data loss?

Line 345-6: Based on the Introduction, it is unclear why this correlation analysis would be performed. Please explain the mechanistic link.

Line 368: For reader convenience, could the HFB vs. LFB distances be shown here?

Line 394-6: Is this outcome surprising? The rise in core temperature is mild.

Line 413-7: This statement is likely correct. Another issue is the variability in the cytokine data that could mask effects.

Line 419-21: Please propose the physiological mechanism explaining the change in core temperature.

6. PLOS authors have the option to publish the peer review history of their article (what does this mean?). If published, this will include your full peer review and any attached files.

Reviewer #1: No

Reviewer #2: **Yes: **Mark Willems

Reviewer #3: No

---

## [Author Response · Author response to Decision Letter 0]

8 Jul 2022

Editor comments 

1) Explain deviation and how it affected the outcomes: inclusion of rather untrained participants. Also, age range is different. 

Based on comment #37 from Reviewer 3 below, we used the classification scheme of De Pauw et al. to characterize our participants, and due to their VO2max (43.2 kg/m2) and power (~229 W) they are considered untrained. 

However, in our opinion, this characterization is misleading. First, the study was conducted in Logan, UT at an approximate elevation of 1500m. According to Wehrlin and Hallen (PMID:16311764), VO2max decreases linearly with increasing altitude, and is about 9% lower at 1500m than at sea level. In addition, VO2max decreases with age. In Table 4.1 of The Physical Fitness Specialist Certification Manual, published by The Cooper Institute for Aerobics Research in Dallas, TX, ranges of VO2max are categorized by age groups into Poor, Fair, Good, Excellent and Superior. If we apply a 9% correction to the VO2max of each participant to adjust for altitude, and then classify their cardiorespiratory fitness by age and gender, nine are classified as Superior, two as Excellent and one as Fair. Thus, we do not feel as though our participant group was untrained when consideration of altitude and age is factored in.

During the study, we did seek approval from our institutional review board to increase the age of participants to 55, and received it. However, this was not clear as we inadvertently uploaded a previously approved version. The more recent version was uploaded with the revised manuscript. 

2) For power: the reasoning for taking 328 as a relevant difference is unclear (the number does not appear in the referred paper)

Author Kung performed the power analysis for her PhD proposal. At the time, we felt the study by Morrison et al. (cited in the manuscript) had the most similar design to the one we planned to conduct. The value 328 ng/ml was estimated based on data provided in that study, and does not appear in the manuscript. Rather, it is the difference in i-FABP between the control and treatment effects. 

3) Full supplement details should be provided, i.e., what was in the 64 g of supplement

The 64g of supplement contained sucrose, maltodextrin, blueberry powder, cocoa powder, whey protein isolate and green tea. For the control, alkalized cocoa powder was substituted for the cocoa powder, and a synthetic blueberry powder for the blueberry powder. The latter was provided by the US Highbush Blueberry Council. 

Reviewer #1

4) My concern is the very tiny sample size, which is computed to be 14, in contradiction to the included protocol, which specifies 20.

When we designed the study, we planned for redundant measures of gut permeability, the lactulose:mannitol differential sugar test, and i-FABP. The n=20 was based on the differential sugar test. In Dr. Kung’s dissertation proposal, which was approved in 2017 by her PhD committee, she estimated 20 subjects would allow for a treatment effect of 257.6 ng/ml in the i-FABP assay. This was adjusted based on the 14 subjects we enrolled on line 277 of the submitted manuscript. We then performed a post-hoc power calculation based on within subject standard deviation. That is presented on line 482 of the previously submitted version. 

While a larger n is always better from a statistical power perspective, and would have likely reduced the post-hoc estimated minimum detectable difference, we do not believe an n of 12 limited our ability to detect a genuine treatment effect, had it existed. It seems clear from Figure 5 in the manuscript that there were no meaningful differences in the post-exercise i-FABP between treatments. 

For previously published studies with i-FABP as the dependent variable after exercise, the majority of published studies we could find had 10 or fewer subjects (7/9). Thus, we do not agree with the characterization of our sample size as tiny. 

5) There needs to be more detail about what the effect size actually means and how the sample size incorporated the crossover design.

The between-subject variance in a parallel design is expected to be larger than the within-subject variance in a crossover. In looking for an estimate of a minimal detectable difference and variation, we found a similar study that also utilized a crossover design. We used the within-subject standard deviation for the change in I-FABP to conduct the power calculation for a quantitative endpoint in a crossover study design.

6) In addition, power is not defined correctly: "the probability is 82% that a significant treatment effect will be detected". You could just say "to achieve 82% power".

Change made

7) With such a small sample size, it calls into question the additional analyses besides the primary, and presumably you would want to adjust the alpha for multiplicities which will reduce the power even more.

As discussed above in comment #1, an n=12 is not a small sample size when compared to studies that have been published with a similar experimental design with similar endpoints. In other studies from our group, we adjust for multiplicities when working with big data sets, such as studies on the microbiome, or metabolomics. However, we have not made such adjustments for secondary measures in a study like this, nor have we noticed it to be common. 

8) Finally, "simple randomization" usually called "complete randomization" is, I guess, tossing a fair coin, which does not ensure equal group sizes, but I couldn't find the group sizes anyway.

Group sizes should be in the CONSORT Figure. More information on the randomization was added at line 167. 

Reviewer #2

9) It is not clear from the title and abstract that the supplementation was a flavonoid/milk drink. Please clarify.

The title and abstract were revised to show that the investigational product was a milk-based flavonoid drink. 

Both the treatment and control were prepared in the same manner with 240 ml 2% fat milk, so the intended comparison was the effect of the added flavonoids. The purpose was to design a realistic/well-rounded pre-workout product that would translate to a “real world” application. A small amount of protein relative to carbohydrate is recommended for pre-exercise nutrition, as it elicits a stronger insulin response and greater glycogen synthesis compared to carbohydrate alone. Pre-workout protein can also increase muscle protein synthesis and post-exercise recovery. Milk is commonly incorporated as a part of exercise nutrition. It has both positive sensory and nutritional (BCAA, protein quality, digestibility) attributes.

10) This also needs to be discussed. There is no mention of milk in the discussion. Could it be that milk affected digestion, absorption and availability of flavonoid-derived metabolites. Please discuss.

It is possible that the milk base may have affected the digestion, absorption, and metabolism of the flavonoids in the HFB. However, any potential flavonoid-milk protein interactions are not likely a limiting factor for flavonoid action in the lower GI tract. Normal digestion and enzymatic processes in the GI tract could potentially digest milk components and disrupt these noncovalent flavonoid-milk protein interactions. Additionally, the primary polyphenol and protein complexes thought to be formed are typically larger molecular weight polyphenols such as tannins rather than the smaller flavonoid molecules. Further discussion is provided in lines 556-574 of the revised manuscript. (Another consideration is that typically flavonoid-milk interactions have not been investigated with HST milk proteins, which may or may not have the same potential to interact with phytocompounds.)

11) It is a concern that baseline IL-6 and TNF-alpha was not detectable, and also not at 4 hr post. This needs to be discussed. In addition, observations 1 hr post were very low. Why is this?

In our experience, and that of colleagues we have spoken to, it is not uncommon for baseline values for inflammatory cytokines to be below the limit of detection for commercial ELISA assays. Author SK performed the analysis following manufacturer’s directions, and the standard curves were acceptable for (i.e. >0.997) but often higher. 

12) The study did not have a full familiarization for the 1 hr cycling test. This needs to be discussed.

The pre-loaded trial design was selected due to its reliability in test-retest designs. In an effort to reduce subject burden, a full 1 hr familiarization trial was not conducted.

13) It is not clear what the cycling distance was during the 15 min TT.

The 15 min TT was an all-out effort, with cycling distance as a dependent variable. LFB: 18.9 ± 0.8 km, HFB: 19.3 ± 0.7 km. This is presented on line 373.

14) On several occasions, statements are supported with review papers. I suggest to clarify that it is a review paper or even better provide original sources.

We have provided more specific information from references in both the Introduction and Discussion. 

15) Please clarify in the abstract whether the 15 min of cycling was subsequent to the 1 hr of cycling, see L112.

The abstract was edited at line 40 to clarify when the time trial was conducted. 

16) In the discussion section of the abstract, it seems the two statement provide similar information. In addition, I suggest to remove the word “significantly” and rephrase “performed as well” suggesting as this suggests a control effect. 

This was edited according to the suggestion at line 54. 

17) In the introduction there is on many occasions mention of polyphenol effects. I suggest to be specific what kind of polyphenols were providing the effect. 

Specific effects of polyphenols were added starting at line 82

18) In the introduction, there is no mention of the flavonoid composition of the supplement of interest and why that composition would have an effect on GI health. The introduction could be strengthened by incorporating studies with effects observed by the flavonoid components of the supplement. 

Information on the effects of blueberry, cocoa and green tea was added starting at line 82. These have all been in preclinical models, which was explicitly stated. The effects of other polyphenols in humans was expanded starting at line 92. 

19) L149. The incremental cycling test started at 100 W and increased 25W/min. For a participant, the power at 70% was 105 W suggesting that this person and maybe more did only a few minutes of cycling to reach VO2max. Please clarify. 

The incremental cycling test started at 100W and proceeded for 10m and resulted in a measure of 41.4 ml/kg/min for the subject’s VO2max. However, we did not base the 70% power used in the 45m cycling trial on this value on the suggestion of author, EB. Rather, we conducted a second test to validate the power at 70% VO2max, as described starting on line 154 of the originally submitted manuscript. In summary, after a 10m rest, participants rode at a steady state and power was adjusted in 5W increments to achieve 70% of the measured VO2max for 3 minutes. In this second test the value of 105 W was determined. 

20) 169. Was 2% the fat content. Please clarify. 

This was clarified on line 226

21) L170. The supplement was dissolved in milk so there was protein intake as well and not only flavonoids. Please clarify. 

We edited the manuscript to state that the product contained protein and fat on line 231.

22) Ls 175-179. Please provide justification for the amounts of green tea extract, cocao and blueberry. 

This was addressed starting at line 216

23) Ls 216-217. What was the composition of the “typical pre-race breakfast”. Why was this not standardized. This needs to be discussed. 

Participants recorded their meals in a food log and replicated them during the second arm. Pre-race breakfast was not standardized due to the study design/objective and study being conducted in free-living subjects.

24) Ls 325. Please provide pre-exercise IL-6 value to allow the statement that there was an increase post-exercise. 

As discussed in the response to comment #11 above, values at this timepoint were below the detection limit of the assay. While we do have values for this timepoint, the fact that were detectable after the ride indicates to us they was an increase post exercise. 

25) L 372. Mention of blackcurrant and pomegranate studies and then the subsequent statement provide observations from a cherry study whereas the “For example,…” would suggest an example of a blackcurrant or pomegranate study. 

This has been edited to differentiate the black currant and pomegranate results from the cherry results. 

26) L395. Why were some so low (and even non-detectable). Was there an issue with the assay? 

We do not believe there was an issue with the assay, as we followed the instructions from the manufacturer, and the standard curves were acceptable. It has been our experience that inflammatory cytokines are not always detected in a relatively healthy population absent a stressor. 

Reviewer #3 

27) Line 64: “high intensity” is missing a hyphen. Please add. 

Change made

28) Line 73: A determiner is missing before “magnitude”. Please add an article. 

Change made

29) Line 85: Please a comma before and after “as determined by plasma I-FABP”. 

Change made

30) Line 93: The preposition use may be incorrect here. Please consider if “at” should be ‘on’. 

In making other edits, this phrase was removed. 

31) Line 100: A determiner is missing before “onset”. Please add an article. 

Change made

32) Line 113: Please consider if “primary endpoint” is the best word choice. ‘primary outcome measure’ might be clearer. 

Change made

33) Line 113: A comma is missing after “injury”. Please add. 

Change made

34) Line 116: Intestinal damage is most often higher in runners than cyclists. With this in mind, why was cycling chosen as the exercise model? 

Similar cycling protocols (~1 h at 70% VO2max) have reliably demonstrated elevated intestinal injury. Additionally, it was more feasible for our lab/clinic to recruit a group of cyclists that could complete the challenging exercise protocol in its entirety than an equivalent group of runners with a running protocol. 

35) Line 124: Please add the ethics approval number. 

This is listed on line 150

36) Line 130: Please reference the tool used to complete the randomization.

This is described on line 169

37) Line 135-6: For ease of comparison with other studies, please consider adopting the descriptors of De Pauw et al. Reference: De Pauw, K., Roelands, B., Cheung, S. S., De Geus, B., Rietjens, G., & Meeusen, R. (2013). Guidelines to classify subject groups in sport-science research. International Journal of Sports Physiology and Performance, 8(2), 111-122.

Revised. Our cyclists fall in the PL 1 category (out of 5) since VO2max < 45, aka “untrained” per De Pauw et al.

38) Line 143: The range in participant characteristics is large. Does this heterogeneity concern the authors? 

Intestinal permeability and the post exercise inflammatory response are of interest to all amateur athletes, regardless of age. While our study did have a large age range, an objective measure of participant fitness, corrected for age, indicated 11/12 had either excellent or superior cardiorespiratory fitness. The exercise trial was tailored to each individual to produce a similar level of stress. The change in core temperature is shown in Figure 4. While there was a curious difference in the baseline, the highest values were all within a 1°C range. To us, this suggests the exercise challenge was similar across all participants. 

We did note that some studies with highly trained cyclists report lower between subject variation in measures of either gut permeability, or inflammation. However, this does not always seem to be the case. Also, when a very specific population of subjects is used, the results theoretically only represent that population. 

39) Line 150: How was a plateau defined? 

We used the definition of Yoon, Kravitz, and Robergs (PMID: 17596788). The slope of the VO2-time value <0.05 L/min. 

40) Line 170-1: What does the subjects’ typical exercise look like? These details are not included in S1 Table. 

We did not ask participants for information on their typical workouts. However, in the Informed Consent document we listed as a requirement for the study that they train at least 3 times per week for one hour on average. In the original manuscript it was not clear if this information was provided in Supplement Table 1, which it is not. The revised manuscript was edited at line 225 to clarify this.

41) Line 171-2: Were the supplements taste matched? Were participants able to detect whether they were in the high or low condition? Does this matter? 

The supplements were matched to the best of our ability. The primary flavor was from the cocoa powder and blueberry powder. For the cocoa, we substituted alkalized cocoa, and for the blueberry powder we used a placebo powder provided by the US Highbush Blueberry Council that was designed to be used as a control. 

42) Line 217-8: Calibration of core temperature pills is ideal (Hunt et al., 2017). Was this undertaken here?

Reference: Hunt, A. P., Bach, A. J., Borg, D. N., Costello, J. T., & Stewart, I. B. (2017). The systematic bias of ingestible core temperature sensors requires a correction by linear regression. Frontiers in Physiology, 8, 260.

We did not calibrate the core temperature pills. The manuscript cited above was published just as we began our study, and we were not aware of it. However, our data are consistent with other exercise studies in terms of the max temperature reached. 

43) Line 218-9: Was hydration examined via either urine or blood? 

We did not examine urine or blood for hydration levels. However, we did provide explicit instructions for rehydration after the exercise trial in the Informed Consent document which are reproduced below. 

You will be advised to rehydrate with 1.5 times the weight you lost in perspiration, and asked to collect your urine for the next 6h. Please repeat the same lunch from the second screening visit and Day 13; we ask that this is all you eat until the end of the urine collection. We will take additional blood draws at 1 and 4 hrs following the exercise challenge.

44) Line 228-9: Please specifically identify the RPE scale used (e.g., Borg’s 6-20 or CR10?). Other details including familiarization and anchoring (Haile, Gallagher & Robertson, 2016) and the standardization of the recording context (Minett et al., 2021), would also be worthwhile.

References: Haile, L., Gallagher, M. J., & Robertson, R. J. (2016). Perceived exertion laboratory manual. Springer-Verlag New York.

Minett, G. M., Fels-Camilleri, V., Bon, J. J., Impellizzeri, F. M., & Borg, D. N. (2021). Peer presence increases session ratings of perceived exertion. International Journal of Sports Physiology and Performance, 17(1), 106-110. 

We revised the manuscript at line 293 to indicate it was Borg’s CR10. We explained the scale to participants, but did not do any familiarization or anchoring. This was evaluated to compare the perceived difficulty of the two exercise trials.

45) Line 238: Could the authors please detail the environmental conditions? 

The cycling was conducted at room temperature and ambient relative humidity in the Center for Human Nutrition Studies, which was consistently 23±1°C. This was added to line 283 of the revised manuscript. We did not measure relative humidity, but the climate in Utah is high desert, and rooms on the Utah State University campus are typically within the range of 20%-50%. 

46) Line 242: How long were samples stored before analysis? 

The samples were stored for 6-18 months in a -80 C (no freeze thaw cycles). The first samples were collected in November of 2018, and the last analyses done in May 2020.

47) Line 247: Please describe the reliability of these measures. 

Author SK conducted these analyses, and followed the manufacturer’s directions. The r2 values of the standards were >0.95, indicating the assay worked correctly. 

48) Line 273: As per the immediately previous comment. 

Based on the text, this comment appears to address the reliability of the urine sugar assay. The standard curves in the GCMS data typically have r2 values >0.999, and replicate analyses of the same sample had coefficient of variances less then 5%. There may be some variance introduced in multiplying the measured concentration by the total urine volume. The recoveries are in line with values from our lab from other studies, and other studies we have found in literature. 

49) Line 279: Is it appropriate to combine male and female data in this context? 

We believe so and did not find studies with similar designs that separated out male and female data. As we discussed in our response to the first comment, after adjusting for age and elevation, 11/12 of our participants had either superior or excellent cardiorespiratory health. Three of the four women rated as superior. We have not come across any information regarding the difference in male and female response of the gastrointestinal barrier to exercise and heat stress. 

50) Line 295: Could the authors justify the presentation of SEM and not SD values? 

In the submission guidelines, we did not see a specific guideline for which to use. We did see the line ‘It should be clear from the text which measures of variance (standard deviation, standard error of the mean, confidence intervals) and central tendency (mean, median) are being presented.’ 

We would be happy to report the data as mean ± SD if the reviewer would prefer this. 

51) Line 318-9: The decline in core temperate after fluid consumption suggests the pills were still in the stomach. This should be acknowledged as stomach temperature, not gastrointestinal temperature, is being reported. 

We instructed the subjects to consume the thermometer pill 2h prior to the exercise bout, as reports in the literature suggested this would have promoted its passage to the small intestine. The possibility it was still in the stomach was added to the manuscript at line 391. We do not believe we have enough information to state this conclusively. 

52) Line 342: “condition” should be in the plural form. Please correct. 

Change made

53) Line 344-5: Did the authors consider using a mixed-method or similar instead of an ANOVA to accommodate this data loss? 

We analyzed the data with the pre-determined statistical analysis plan and used a paired t-test.

54) Line 345-6: Based on the Introduction, it is unclear why this correlation analysis would be performed. Please explain the mechanistic link. 

We included this correlation analysis because changes in gut permeability are associated with the magnitude of core temperature change, according to the Pires reference cited in the manuscript. 

55) Line 368: For reader convenience, could the HFB vs. LFB distances be shown here? 

The data is presented in Figure 3. We would be happy to accommodate this request if we misunderstood the reviewer request. 

56) Line 394-6: Is this outcome surprising? The rise in core temperature is mild. 

We believe the change in cytokines was consistent with the change in core temperature. 

57) Line 413-7: This statement is likely correct. Another issue is the variability in the cytokine data that could mask effects. 

This is an interesting point, but it is unclear if the reviewer would like us to explicitly state this in the Discussion. 

58) Line 419-21: Please propose the physiological mechanism explaining the change in core temperature.

This was added at line 497 in the revised manuscript.

---

## [Decision Letter · Decision Letter 1]

8 Aug 2022

PONE-D-22-01325R1No effect of a dairy-based, high flavonoid pre-workout beverage on exercise-induced intestinal injury, permeability, and inflammation in recreational cyclists: A randomized controlled crossover trialPLOS ONE

Dear Dr. Ward,

Thank you for submitting your manuscript to PLOS ONE. After careful consideration, we feel that it has merit but there are a few minor points that need additional attention. Therefore, we invite you to submit a revised version of the manuscript that addresses the points raised during the review process.

Please consider the additional comments Reviewer 2 made regarding the warmup protocol and clarification on the TT data. ==============================

We look forward to receiving your revised manuscript.

Kind regards,

Lex Verdijk, PhD

Academic Editor

PLOS ONE

Journal Requirements:

Reviewers' comments:

Reviewer's Responses to Questions

**Comments to the Author**

1. If the authors have adequately addressed your comments raised in a previous round of review and you feel that this manuscript is now acceptable for publication, you may indicate that here to bypass the “Comments to the Author” section, enter your conflict of interest statement in the “Confidential to Editor” section, and submit your "Accept" recommendation.

Reviewer #1: All comments have been addressed

Reviewer #2: (No Response)

Reviewer #3: All comments have been addressed

2. Is the manuscript technically sound, and do the data support the conclusions?

Reviewer #1: (No Response)

Reviewer #2: Yes

Reviewer #3: Yes

3. Has the statistical analysis been performed appropriately and rigorously? 

Reviewer #1: (No Response)

Reviewer #2: Yes

Reviewer #3: I Don't Know

4. Have the authors made all data underlying the findings in their manuscript fully available?

Reviewer #1: (No Response)

Reviewer #2: Yes

Reviewer #3: Yes

5. Is the manuscript presented in an intelligible fashion and written in standard English?

Reviewer #1: (No Response)

Reviewer #2: Yes

Reviewer #3: Yes

6. Review Comments to the Author

Reviewer #1: (No Response)

Reviewer #2: L157. There is no mention, according to your reply, that 100 W was cycled at for 10 min before initiation of the increments. Please revise.

L331. It seems to be stated here that the distance covered during the 15 min TT was more than 18 km, but total distance is reported. Please provided the correct distances for the TT only. See also Figure 3 legends that requires change.

L333. Are the work data also from the 60 min cycling. If that is the case, I suggest to present the TT work data.

Reviewer #3: Thank you for making these changes. It is not easy to accommodate the requests of multiple reviewers.

7. PLOS authors have the option to publish the peer review history of their article (what does this mean?). If published, this will include your full peer review and any attached files.

Reviewer #1: No

Reviewer #2: **Yes: **Mark Willems

Reviewer #3: No

---

## [Author Response · Author response to Decision Letter 1]

21 Sep 2022

Reviewer #2: L157. There is no mention, according to your reply, that 100 W was cycled at for 10 min before initiation of the increments. Please revise.

Yes, 100 W was not cycled at for 10 min before initiation of the increments. All subjects followed the same pre-defined VO2max test and steady state validation assessment protocol. After a 5-minute warm-up at 100 W, the test started at 100 W, increasing 25 W/min, and ended when volitional exhaustion, rpm < 60, or a plateau in VO2 was observed (when a VO2 measurement was followed by two consecutive values lower than the first, despite continual effort by the subject). After an initial 5 min at 100 W, this subject cycled for 5 minutes during their VO2max test and finished at 225 W. The subject's 70% VO2max was 28.98, and after the second steady state validation assessment, the final power at 70% VO2max was determined to be 105 W. 

We made the following edit at line 155 to clarify this…

After a five-minute warm-up at 100 W, the incremental graded VO2max test began at 100 Watts (W) and increased by 25 W/min until either volitional exhaustion, rpm < 60, or a plateau in VO2 was observed.

L331. It seems to be stated here that the distance covered during the 15 min TT was more than 18 km, but total distance is reported. Please provided the correct distances for the TT only. See also Figure 3 legends that requires change.

An edit was made to L378 (tracked changes file) to clarify that it is the TT distance reported. We believe the legends for Figure 3 are also clear that this is the TT performance metrics, but would consider any edits the reviewer may suggest to clarify this point. 

L333. Are the work data also from the 60 min cycling. If that is the case, I suggest to present the TT work data.

The TT work data are presented. A minor edit was made to clarify on L380 of tracked changes file.

---

## [Editor Report · Decision Letter 2]

30 Sep 2022

PONE-D-22-01325R2No effect of a dairy-based, high flavonoid pre-workout beverage on exercise-induced intestinal injury, permeability, and inflammation in recreational cyclists: A randomized controlled crossover trialPLOS ONE

Dear Dr. Ward,

Thank you for submitting your manuscript to PLOS ONE. As it appears, I think that the authors did not truly understand the final comment(s) from reviewer 2 in the last revision round. Hence, below I have provided a more specific comment that needs to be clarified before we can fully accept your work for publication. We invite you to submit a revised version of the manuscript that addresses this point.

In the previous revision round, reviewer 2 mentioned that the data provided in Figure 3 appear to represent the total distance covered over the full 1-hour cycling protocol (so the 45 min at 70%-power, plus the 15 min all-out TT). I understand that point, since the participants were not particularly well-trained average VO2max of 43 ml/kg/min, max power output approximately 225 @). With such a training status it seems impossible to cycle 19 km in 15 minutes!! Also, the power output during the 'all-out' 15 min-TT does not seem to be much different from the 70% intensity during the first 45 minutes, and would not allow to be translated to 19 km. The reviewer therefore assumed that the data in figure 3A and 3B are actually the data from the full 60-minutes of cycling rather than only the 15-min TT. Although 19 km covered distance within an hour cycling would be on the low side, the same distance within only 15 min seems impossible. This inconsistency should be clarified. The authors have now responded by stating that these data are actually from the 15-min TT, but then the 19 km covered distance does not make much sense, as they would be cycling at 76 km/h!! Please double-check the calculations for covered distance and power output and report those data only for the 15-min TT in figure 3. 

A marked-up copy of your manuscript that highlights changes made to the original version. You should upload this as a separate file labeled 'Revised Manuscript with Track Changes'.An unmarked version of your revised paper without tracked changes. You should upload this as a separate file labeled 'Manuscript'.If applicable, we recommend that you deposit your laboratory protocols in protocols.io to enhance the reproducibility of your results. Protocols.io assigns your protocol its own identifier (DOI) so that it can be cited independently in the future. For instructions see: https://journals.plos.org/plosone/s/submission-guidelines#loc-laboratory-protocols. Additionally, PLOS ONE offers an option for publishing peer-reviewed Lab Protocol articles, which describe protocols hosted on protocols.io. Read more information on sharing protocols at https://plos.org/protocols?utm_medium=editorial-email&utm_source=authorletters&utm_campaign=protocols.

We look forward to receiving your revised manuscript.

Kind regards,

Lex Verdijk, PhD

Academic Editor

PLOS ONE
---

## [Author Response · Author response to Decision Letter 2]

6 Oct 2022

Reviewer/Editor Comment: 

In the previous revision round, reviewer 2 mentioned that the data provided in Figure 3 appear to represent the total distance covered over the full 1-hour cycling protocol (so the 45 min at 70%-power, plus the 15 min all-out TT). I understand that point, since the participants were not particularly well-trained average VO2max of 43 ml/kg/min, max power output approximately 225 @). With such a training status it seems impossible to cycle 19 km in 15 minutes!! Also, the power output during the 'all-out' 15 min-TT does not seem to be much different from the 70% intensity during the first 45 minutes, and would not allow to be translated to 19 km. The reviewer therefore assumed that the data in figure 3A and 3B are actually the data from the full 60-minutes of cycling rather than only the 15-min TT. Although 19 km covered distance within an hour cycling would be on the low side, the same distance within only 15 min seems impossible. This inconsistency should be clarified.

The authors have now responded by stating that these data are actually from the 15-min TT, but then the 19 km covered distance does not make much sense, as they would be cycling at 76 km/h!! Please double-check the calculations for covered distance and power output and report those data only for the 15-min TT in figure 3. 

Response: 

Somehow, we missed this. The reviewer is correct. The distance was for the whole time on the bike, and not only the time trial. We have corrected this in the resubmitted manuscript. The work reported, however, was correct. These edits do not change the outcome or study findings. 

We appreciate the comments and are a bit embarrassed we missed this the last time.

---

## [Editor Report · Decision Letter 3]

28 Oct 2022

No effect of a dairy-based, high flavonoid pre-workout beverage on exercise-induced intestinal injury, permeability, and inflammation in recreational cyclists: A randomized controlled crossover trial

PONE-D-22-01325R3

Dear Dr. Ward,

We’re pleased to inform you that your manuscript has been judged scientifically suitable for publication and will be formally accepted for publication once it meets all outstanding technical requirements.

Kind regards,

Lex Verdijk, PhD

Academic Editor

PLOS ONE

Additional Editor Comments (optional):

Please carefully check all files and get into contact with the editorial office, as in the latest file I missed the revised figure 3B. I am not sure if this was incorrectly incorporated in the pdf or it was not uploaded correctly. In any case, this is a point to check precisely (with the adjusted distance, i.e. ~7.3 km for the TT) for the final files to be published!
---

## [Editor Report · Acceptance letter]

8 Nov 2022

PONE-D-22-01325R3 

No effect of a dairy-based, high flavonoid pre-workout beverage on exercise-induced intestinal injury, permeability, and inflammation in recreational cyclists: A randomized controlled crossover trial 

Dear Dr. Ward:

I'm pleased to inform you that your manuscript has been deemed suitable for publication in PLOS ONE. Congratulations! Your manuscript is now with our production department. 

Kind regards, 

on behalf of

Dr. Lex Verdijk 

Academic Editor

PLOS ONE